# Study on Effects of Nonlinear Behavior Characteristics of Prepreg Dielectric on Warpage of Substrate under Laminating Process

**DOI:** 10.3390/polym14030561

**Published:** 2022-01-29

**Authors:** Seunghyun Cho, Youngbae Ko

**Affiliations:** 1Department of Mechanical Engineering, Dongyang Mirae University, Seoul 08221, Korea; coolcsh@dongyang.ac.kr; 2Korea Institute of Industrial Technology (KITECH), Jeju 63243, Korea

**Keywords:** nonlinear characteristics, prepreg dielectric, substrate, warpage, stress, FEM

## Abstract

To analyze the effects of nonlinear behavior characteristics of prepreg (PPG) among the insulating materials of substrate and the residual stress of laminating process on the warpage of substrate, this study investigated the continuous laminating process using the numerical analysis by finite element method. The analysis results showed that the warpage of the substrate in the laminating process of PPG was very low, but it increased rapidly in the solder resist (SR) laminating process. As the laminating process of PPG continued, the stress inside the substrate increased continuously and it was predicted to decrease in the SR laminating process. These results confirmed that the warpage of the substrate is influenced the most by the SR laminating process, and that the warpage and stress of substrate accumulated in the laminating process of PPG have significant effects on the final warpage.

## 1. Introduction

As the thickness of the semiconductor substrate decreases and the core disappears, the specific gravity of the insulating material in the substrate increases. At the same time, the increasing deformation of the substrate not only causes defects in the manufacturing process, but also has a low reliability [1,2,3]. The semiconductor substrate repeatedly undergoes heating processes such as prepreg (PPG) laminating, solder resist (SR) laminating, and SR printing. Furthermore, the packaging stage also includes various thermal processes such as chip assembly, wire bonding, flip chip bonding, mold cure, and soldering. Many reliability defects are caused by the low stiffness and warpage of the substrate resulting from thermal processes, such as underfill, detachment and crack of solder joints, and disconnection of the solder joint [4,5]. Thus, the warpage of the substrate is known as the fundamental cause of such reliability issues [6,7,8,9,10,11,12]. In this respect, the effects of the behavior characteristics of insulating materials on the warpage are being continuously researched for four to six layer coreless flip chip substrates [13].

Core or coreless substrates have many restrictions due to their low stiffness, because as the thickness becomes smaller, the stiffness decreases in proportion with the power of thickness. However, in thermal processes, the nonlinear behavior characteristics of insulating materials in the substrate have greater effects than the stiffness of the substrate, making it very difficult to predict the warpage [14,15]. In the semiconductor packaging field, studies on the nonlinear behavior characteristics of materials have been actively conducted, which mainly target the underfill film or molding encapsulant [16,17,18,19]. Recently, however, many studies on nonlinear behavior characteristics of insulating materials in the substrate have been conducted. With the development of various transformation model formulas, studies on the effects of the nonlinear behaviors of materials on warpage have been conducted using simulation technology [20,21,22,23,24,25,26].

The purpose of this paper is to numerically calculate the warpage and stress generated during the continuous PPG and SR laminating processes of PCB. This is because the warpage generated in the laminating process is confirmed, but it is very difficult to measure it quantitatively.

Therefore, to analyze the effects of the nonlinear behavior characteristics of PPG on the warpage of the substrate, this study conducted a numerical analysis using the finite element method for the PPG and SR laminating processes, which are major thermal processes. Furthermore, the warpage and stress that occur during continuous laminating processes were considered so as to examine the conditions of the actual manufacturing site. To that end, the nonlinear behavior characteristics of PPG, such as cure degree, modulus, coefficient of thermal expansion (CTE), shrinkage, and viscoelasticity, were measured, evaluated, and applied to the material characteristic values of the modeling. This study is expected to provide useful information for substrate development and manufacturing engineers in order to understand the warpage patterns before and after mounting semiconductor chips, and to manage processes that minimize the warpage of the substrate.

## 2. Materials and Methods

### 2.1. Evaluation of Material Characteristics

#### 2.1.1. Cure Degree

The cure degree of the insulating materials in the substrate is a major factor that influences warpage because the materials shrink in the curing reaction. Figure 1 shows the analysis results of the cure degree of PPG. In this study, the cure degree of PPG was measured according to temperature, under the condition that the temperature was increased at the rate of 5 °C/min in the DSC(differential scanning calorimetry; TA instrument, New Castle, DE, USA) using PPG 12.3 mmg of Mitsubishi, including glass fibers, and then the same temperature was maintained. After approximately 250 min, the cure degree was measured as approximately 91.3% at 200 °C, 94.7% at 210 °C, 96.7% at 220 °C, 97.9% at 230 °C, and 98.7% at 240 °C. In general, products with a cure degree of 95% or higher were used, and it was considered that the appropriate curing temperature was 220 °C or higher.

#### 2.1.2. Storage Modulus

Figure 2a,b shows the measurement results of the storage modulus of PPG. Specimens with a width × length × thickness of 3.28 mm × 50.2 mm × 0.065 mm were measured using DMA(dynamic mechanical analyzers; TA instrument, New Castle, DE, USA) in 1 Hz tension mode. After the temperature increased from the room temperature to 260 °C at a rate of 5 °C/min, it was maintained for 60 min and then decreased to the room temperature at the same rate, before the temperature was increased to 300 °C at the same rate. Figure 2a shows the modulus in the x direction. In Section ①, this figure is before the curing reaction of PPG, and the modulus decreased due to the softening of the material. Then, in Section ②, the modulus increased near approximately 150 °C when the curing reaction started. In Section ③, where the temperature was maintained, the modulus continuously increased due to the curing reaction of PPG. In Section ④, the temperature was decreased to room temperature and the modulus increased due to cooling. In Section ⑤, where the temperature rose again, the modulus decreased. The sections of the modulus before full curing are ① and ②, and the sections of the modulus after full curing are ④ and ⑤. In Sections ④ and ⑤ after full curing of PPG, the modulus was measured to be higher than before full curing, and the modulus stayed constant according to the change of temperature. Figure 2b shows the storage modulus measured in the y direction, which was higher than in the x direction, and its change pattern according to the temperature was the same as in the x direction.

#### 2.1.3. CTE (Coefficient of Thermal Expansion)

Figure 3 and Figure 4 show the CTE measurement results of PPG. A specimen with a width × length × thickness of 3.1 mm × 7.3 mm × 0.065 mm was measured in 0.05 N tension mode using TMA(thermomechanical analyzers; TA instrument, New Castle, DE, USA). After the temperature was increased from room temperature to 260 °C at a rate of 5 °C/min, which was maintained for 10 min, the dimensional change was measured during natural cooling back to room temperature.

Figure 3 shows the dimensional change in the x direction. To calculate the CTE accurately according to the temperature change, Section ① with a large dimensional change during the temperature rise was subdivided into 11 subsections. However, Section ②, where the dimension changed constantly while the temperature decreased, was not subdivided. Table 1 shows the CTE in each temperature section in the x direction calculated in this way. The equation for CTE is as follows:(1)$$\mathrm{CTE}=\frac{\left(Dim.2-Dim.1\right)}{\left(Temp.2-Temp.1\right)\times 0.0073m}$$

Figure 4 shows the results of measuring the dimensional change in the y direction, which was smaller than that in the x direction. To calculate the CTE, Section ① was subdivided into eight subsections while the temperature rose, but Section ② was not subdivided. Table 2 shows the CTE in each temperature section in the y direction.

#### 2.1.4. Cure Shrinkage

Transformation by CTE did not occur if there was no temperature change (∆T). Thus, there was no transformation by CTE if the temperature returned to room temperature after rising from room temperature. However, as shown in Figure 3 and Figure 4, a residual transformation of the PPG occurred even after the temperature returned to the initial temperature after rising. In this case, it could be assumed that the transformation was caused by the cure shrinkage of PPG.

Figure 5 shows the cause of dimensional change of PPG. When the temperature rose from the initial temperature, $T_{0}$, to the highest temperature, $T_{2}$, and then returned to $T_{0}$, the dimensional change increased from $\;\delta_{0}$ to $\;\delta_{2}$ and then returned to $\;\delta_{0}$. Here, the cause of dimensional change $\;\delta_{0}-\delta_{2}$ was the CTE of PPG. However, the actual final dimensional change of PPG is $\;\delta_{3}$, and it is considered that the cause of $\;\delta_{0}-\delta_{3}$ is not CTE, but cure shrinkage. For this reason, the cause of the residual dimensional change after the temperature, which returned to room temperature in Figure 3 and Figure 4, could be said to be the cure shrinkage.

Figure 6 shows the ratio of volume change in the PPG measured under the same conditions as for the CTE measurement. The ratio of volume change was calculated by the product of the numerical change in the x and y directions under the assumption that there was no numerical change along the thickness. In this way, the ratio of volume change was calculated as approximately 17.8% and the ratio of volume change by the cure shrinkage was calculated as approximately 0.4%. These results indicate the total volume change occurred for a very short time of 20~30 min.

#### 2.1.5. Viscoelasticity

When a shear strain is applied to an elastic body, the stress is generated in the same form as that of the shear strain in general, but the viscoelastic material is generated as the stress is delayed. Here, the part generated in the same form is called the “storage modulus” and the part generated with a delay is called the “loss modulus”. Thus, storage modulus means the energy stored without loss by elasticity, and loss modulus means the energy lost by viscosity. PPG is a composite structure based on an epoxy in which fibers are impregnated, and thus requires consideration of the effect of the epoxy’s viscoelastic characteristics. In this study, the viscoelastic characteristics of the two directions of x and y were evaluated using DMA(dynamic mechanical analyzers; TA instrument, New Castle, DE, USA). Under tensile conditions of 1, 3, 5, 10, and 15 Hz, the storage modulus and loss modulus of PPG were evaluated in the temperature range of 30 to 260 °C.

Figure 7a,b shows the measurement results of the storage modulus and loss modulus in the x direction of PPG before full curing. They decreased as the temperature increased, but began to increase from 150–160 °C when the curing reaction started.

Figure 8a,b shows the measurement results in the x direction of PPG after full curing. As the temperature increased, the storage modulus slowly decreased and the loss modulus slowly increased.

Figure 9 and Figure 10 show the measurement results of storage modulus and loss modulus in the y direction before and after full curing.

After evaluating the storage modulus and loss modulus as described above, the master curve showing the viscoelastic behavior characteristics must be derived as a function of frequency and temperature. In this study, the master curve was derived by applying the Williams Landel Ferry (WLF) shift function. This is expressed as follows, where the material constants $C_{1}$ and $C_{2}$ were calculated using the commercial program MSC/MARC2019(mscsoftware, Irvine, CA, USA) [27].
(2)$${\log\alpha }_{T}=-\frac{C_{1}\left(T-T_{0}\right)}{C_{2}+\left(T-T_{0}\right)}$$

$C_{1}$: material constants

$C_{2}$: material constants

$T_{0}$: reference temperature, 25 °C
(3)$$\alpha_{T}=\frac{\rho \left(T\right)}{\rho \left(T_{0}\right)}$$

$\alpha_{T}$: shift factor

ρ: relaxation time

The calculated material constants $C_{1}$ and $C_{2}$ are outlined in Table 3.

The master curves derived in the x and y directions of PPG by the WLF shift function are shown in Figure 11 and Figure 12. The master curves were expressed by the modulus according to the frequency of the log scale. It can be seen that the master curve after the full curing of PPG was highly stable according to frequency and temperature compared to the curing process.

### 2.2. Finite Element Analysis

#### Modeling

The warpage of the substrate according to the curing characteristics and viscoelastic characteristics of the PPG insulating materials was analyzed in this study using the three-step laminating process and SR process conditions, as shown in Figure 13. For this analysis, the changes of material characteristics during the processes were assumed. In the first laminating, the PPGs were laminated without curing. In the second laminating, the PPGs that were not cured in the first laminating were laminated and the uncured PPGs were newly laminated. This process was repeated in the third laminating. After this, in the SR process, it was assumed that all PPGs were cured and the SR (film type) condition was cured as well.

Figure 14 shows a diagram illustrating finite element modeling. The substrate that consisted of dummy and unit areas had a four-layer circuit, and only a 1/4 form was modeled using a hexahedral mesh with eight nodes. The number of nodes and hexagonal elements were 124,977 and 95,532 respectively. To analyze the effects of the laminating process and the curing and viscoelasticity of PPG on the warpage of the substrate in this study, the press for laminating was modeled as a rigid body surface (RBS) and the substrate as a deformable body. For the finite element analysis, the thermo-mechanical coupled contact was analyzed using the commercial program MSC/MARC2019. The material properties used in this analysis are listed in Table 4 [28].

Figure 15 shows the analysis conditions according to the laminating process for the finite element analysis. For lamination, RBS was heated while the substrate was pressed, and the pressed amount was 5 μm. After this process was repeated from the first to the third laminating, natural cooling to 25 °C was assumed after heating without pressing in the SR laminating process. The temperature conditions of the PPG and SR laminating processes are shown in Figure 16.

Furthermore, to reflect the effects of the laminating process, the results of the first laminating analysis were entered as the initial conditions of the second laminating analysis, the results of the second laminating analysis were entered as the initial conditions of the third laminating analysis, and the results of the third laminating analysis were entered as the initial conditions of the SR laminating analysis.

## 3. Results and Discussion

Figure 17a,b shows the analysis results for the warpage and stress distributions that occurred in the substrate immediately after the first PPG laminating process and the removal of the press compression for laminating. A warpage of approximately 14.73 μ occurred in the perimeter of the substrate, and the stress by compression was approximately 1.39 MPa.

The warpage and stress distributions that occurred in the substrate after completion of the second PPG lamination with this warpage and stress reflected as the initial conditions are shown in Figure 18a,b, respectively. The warpage was 14.72 μm, which hardly changed, but the stress increased to 38.7 MPa.

The warpage and stress distributions that occurred in the substrate after completion of the third PPG lamination with the above warpage and stress reflected as the initial conditions are shown in Figure 19a,b, respectively. The warpage decreased somewhat to 11.01 μm and the stress increased significantly to 70.57 MPa. These results suggest that as the PPG laminating process was repeated, the warpage decreased slightly, but the stress accumulated inside the substrate.

Figure 20a–c shows the warpage and stress distributions that occurred in the substrate when the SR laminating process was completed, and the substrate was naturally cooled to 30 °C after the warpage and stress in the third laminating process were entered as the initial conditions. Due to the effect of the deformation of SR with a large CTE, the warpage increased rapidly to 10.06 mm. However, as with the warpage, the stress that accumulated inside the substrate during the PPG laminating process decreased to approximately 65.7 MPa. The stress distribution in Figure 20c shows that a large stress occurred in the dummy area outside the unit area.

Figure 21a–c shows the warpage and stress distributions that occurred in the substrate after the SR laminating process was completed, and the substrate was naturally cooled to 30 °C without entering the warpage and stress that occurred in the third laminating process as the initial conditions in order to analyze the effects of the warpage and stress during the laminating process on the final warpage. As shown in this result, when the warpage and stress accumulated in the PG laminating process was not reflected, the final warpage decreased significantly to 0.42 mm and the stress also decreased significantly to 42.3 MPa. The warpage was distributed evenly, without differences between the unit and dummy areas for the stress distributions in Figure 20c.

The results in Figure 20 and Figure 21 suggest that it is necessary to manage the warpage and stress that occur in the laminating process, that is, to develop a method of decreasing warpage and stress in order to reduce the final warpage of the substrate. These results are in close agreement with recent simulation results on two-dimensional layers of colloids in which cooling−heating cycles reduce the stress in the material [29,30].

## 4. Conclusions

In this paper, numerical analysis was performed in consideration of the nonlinear behavior characteristics of materials in order to quantitatively analyze the warpage and stress generated in the continuous laminating process of PPG and SR.

To analyze the effects of the curing and viscoelastic characteristics of PPG on the warpage of the substrate, the continuous laminating process was investigated using numerical analysis using the finite element method. To that end, the elastic modulus and CTE were measured and the volume change ratio and viscoelastic characteristics were calculated and reflected in the analysis. According to the analysis results, the warpage in the laminating process of PPG was very low, but it increased sharply in the SR laminating process. Furthermore, it was predicted that the stress inside the substrate continuously increased during the PPG laminating process and decreased during the SR laminating process. These results confirmed that the warpage of the substrate was most influenced by the SR laminating process. Furthermore, it was found that the warpage and stress of the substrate accumulated in the PPG laminating process had a significant effect on the final warpage of the substrate. Therefore, it is necessary to research methods to reduce the warpage in the laminating process and the residual stress of the substrate.

## Figures and Tables

**Figure 1 polymers-14-00561-f001:**
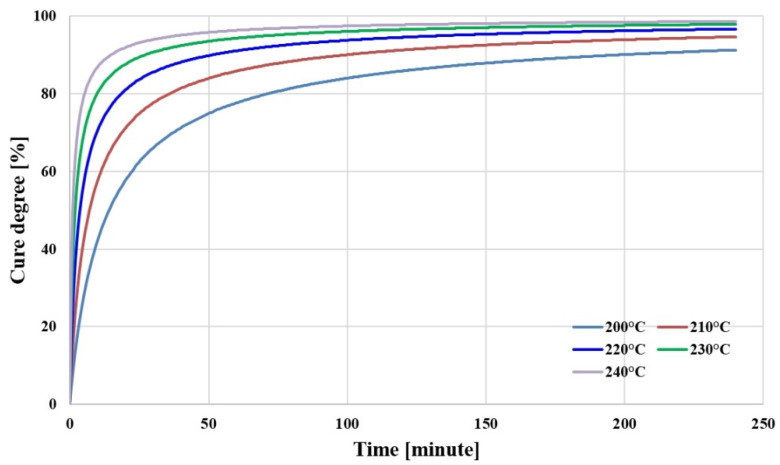
Cure degree of PPG as a function of temperature and time.

**Figure 2 polymers-14-00561-f002:**
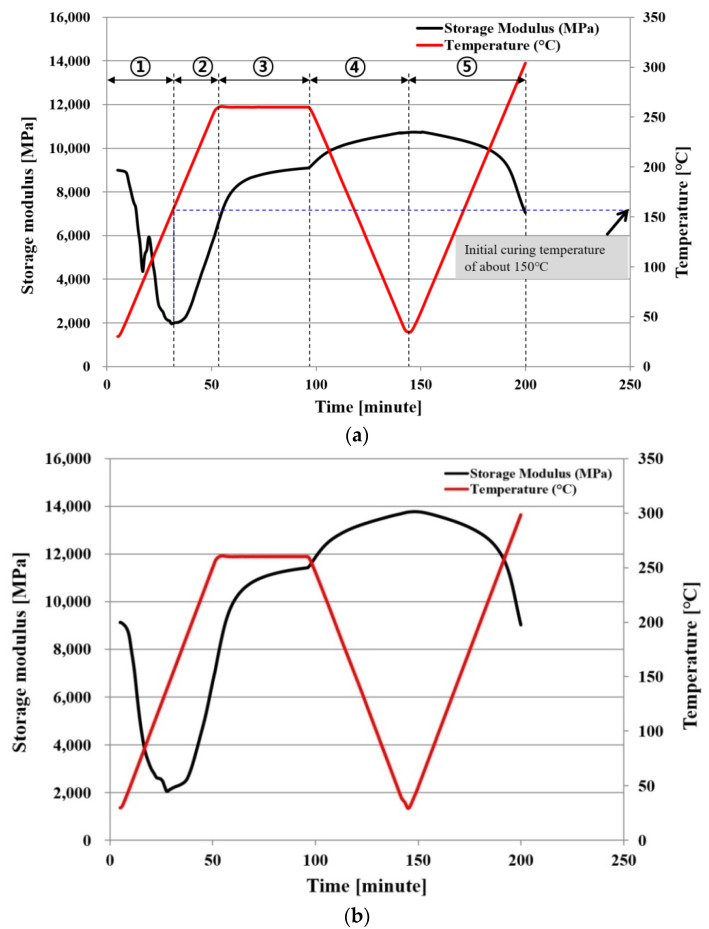
Storage modulus of PPG as a function of temperature and time: (**a**) x direction; (**b**) y direction.

**Figure 3 polymers-14-00561-f003:**
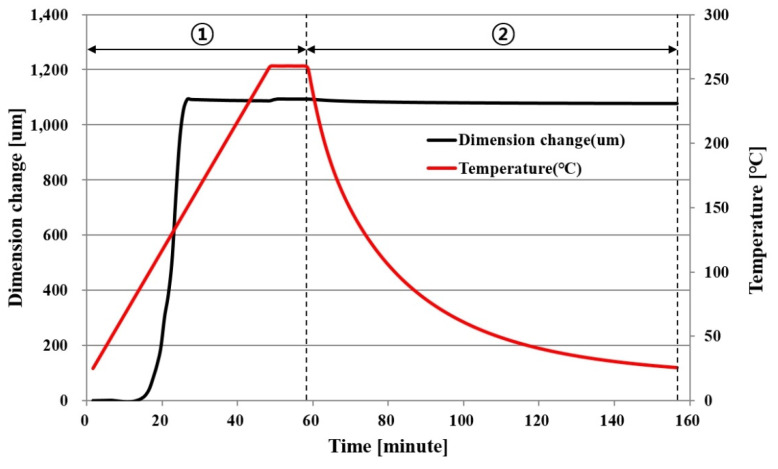
CTE of PPG in an x direction as a function of temperature and time.

**Figure 4 polymers-14-00561-f004:**
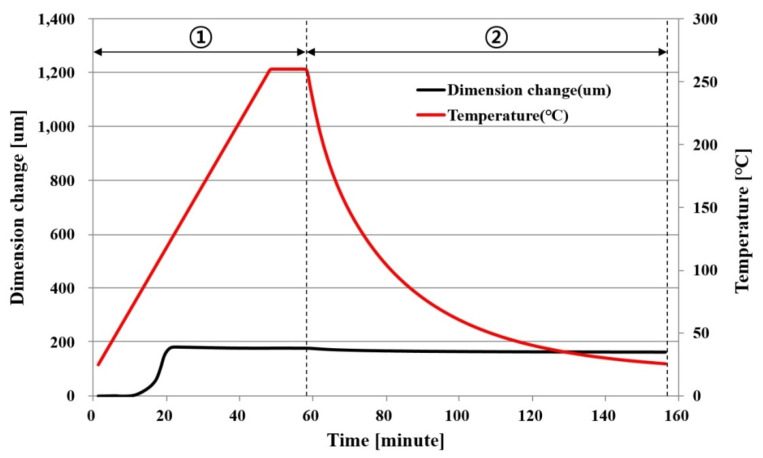
CTE of PPG in a y direction as a function of temperature and time.

**Figure 5 polymers-14-00561-f005:**
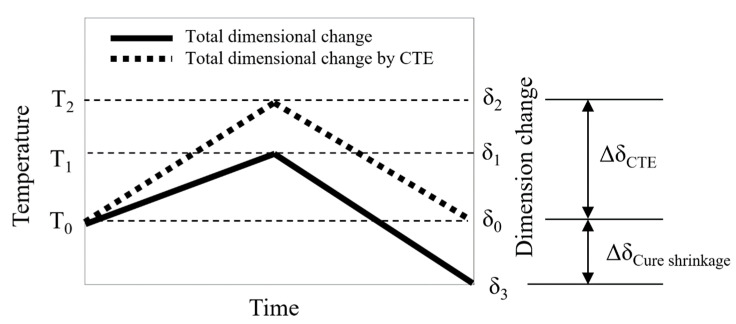
Dimension change by CTE and cure shrinkage.

**Figure 6 polymers-14-00561-f006:**
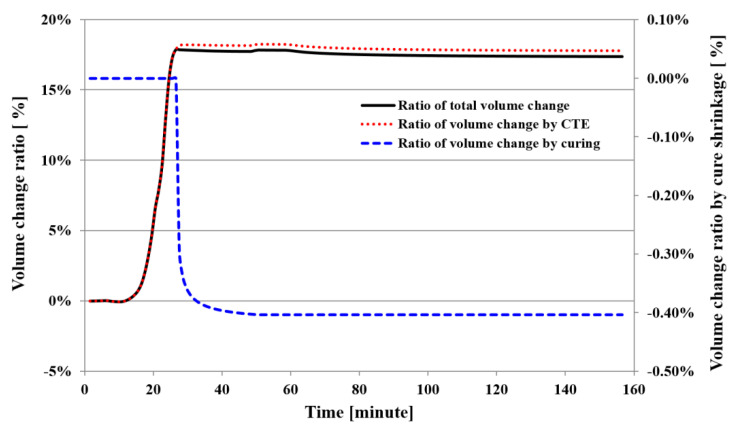
Volume change ratio of PPG as a function of temperature and time.

**Figure 7 polymers-14-00561-f007:**
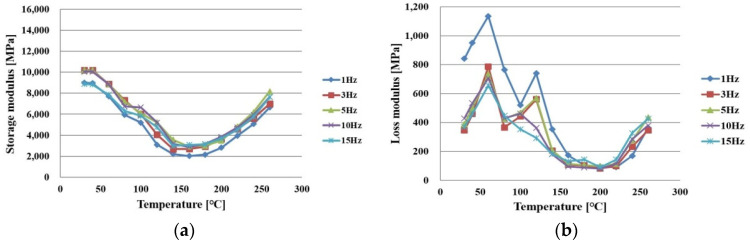
Measurement results of PPG in an x direction under curing as a function of frequency: (**a**) storage modulus; (**b**) loss modulus.

**Figure 8 polymers-14-00561-f008:**
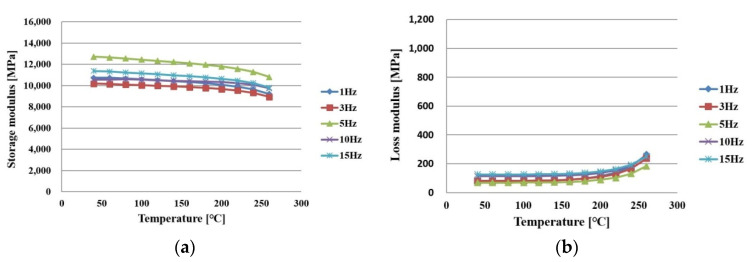
Measurement results of PPG in an x direction after being cured as a function of frequency: (**a**) storage modulus; (**b**) loss modulus.

**Figure 9 polymers-14-00561-f009:**
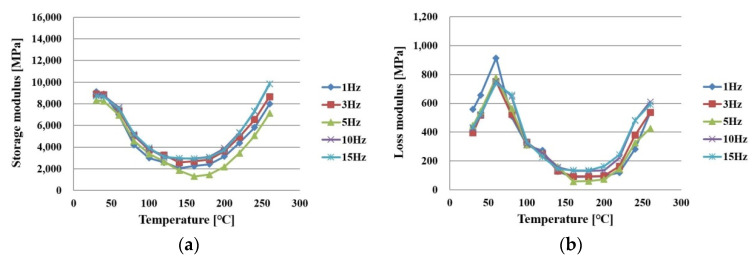
Measurement results of PPG in a y direction under curing as a function of frequency: (**a**) storage modulus; (**b**) loss modulus.

**Figure 10 polymers-14-00561-f010:**
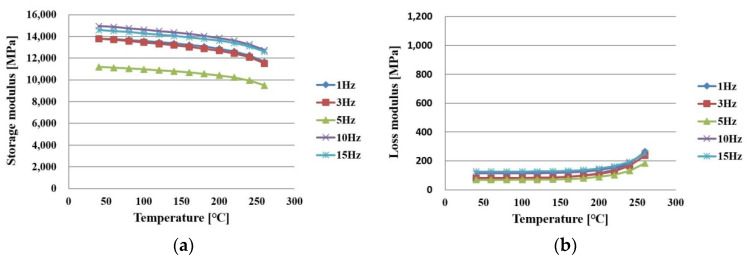
Measurement results of PPG in a y direction after being cured as a function of frequency: (**a**) storage modulus; (**b**) loss modulus.

**Figure 11 polymers-14-00561-f011:**
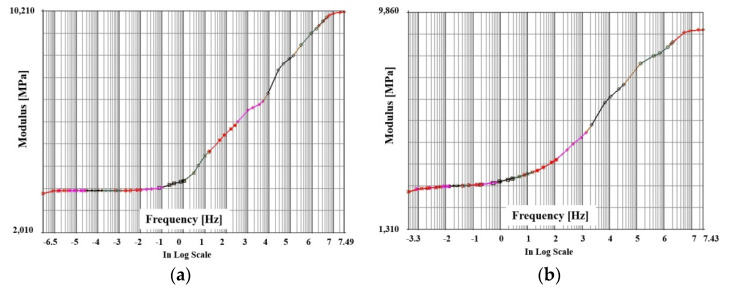
Master curve of PPG under curing as a function of frequency: (**a**) x direction; (**b**) Y direction.

**Figure 12 polymers-14-00561-f012:**
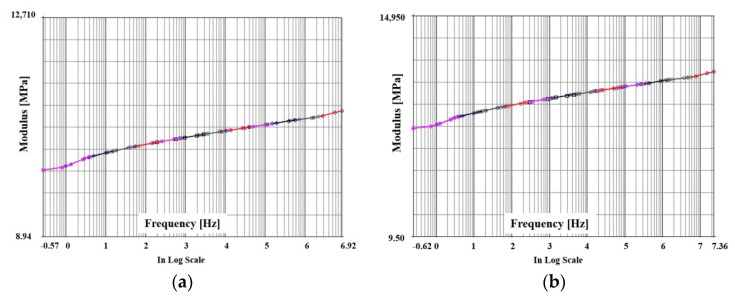
Master curve of PPG after cured as a function of frequency: (**a**) x direction; (**b**) y direction.

**Figure 13 polymers-14-00561-f013:**
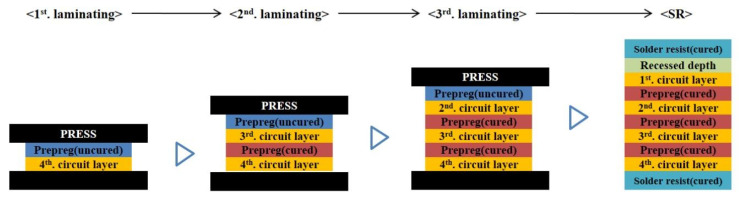
PPG and SR laminating process.

**Figure 14 polymers-14-00561-f014:**
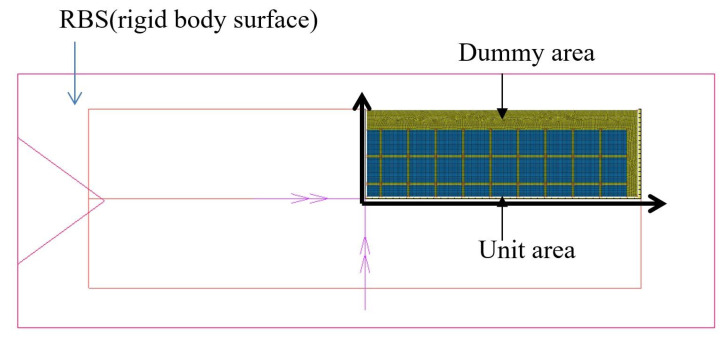
Finite element modeling.

**Figure 15 polymers-14-00561-f015:**
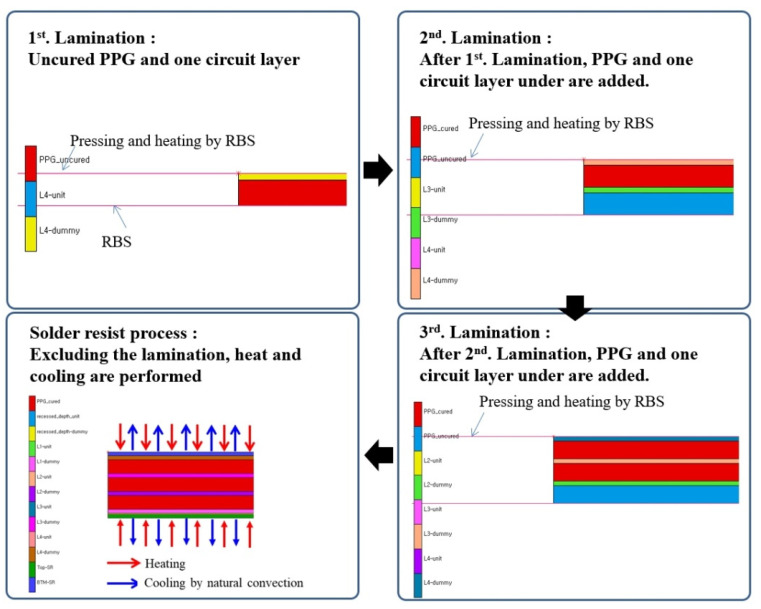
Boundary conditions of the laminating process.

**Figure 16 polymers-14-00561-f016:**
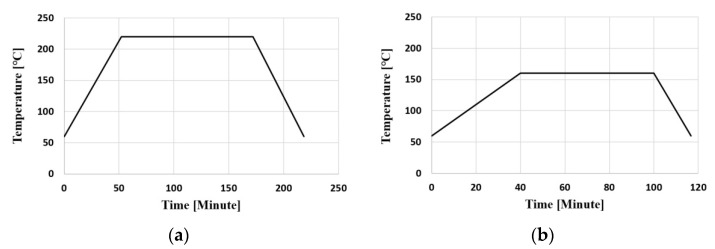
Laminating temperature as a function of time: (**a**) PPG; (**b**) SR.

**Figure 17 polymers-14-00561-f017:**
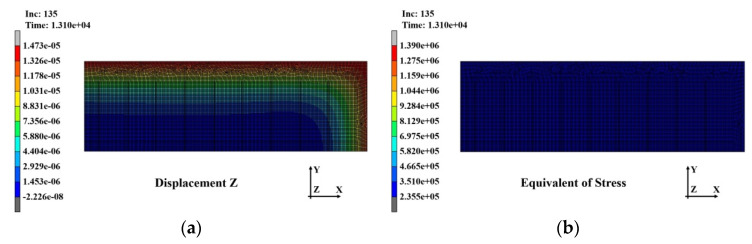
Analysis results of the substrate after first^.^ laminating: (**a**) warpage distribution (14.73 μm); (**b**) stress distribution (1.39 MPa).

**Figure 18 polymers-14-00561-f018:**
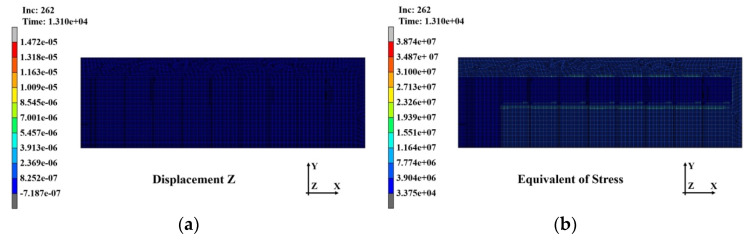
Analysis results of the substrate after second^.^ laminating: (**a**) warpage distribution (14.72 μm); (**b**) stress distribution (38.7 MPa).

**Figure 19 polymers-14-00561-f019:**
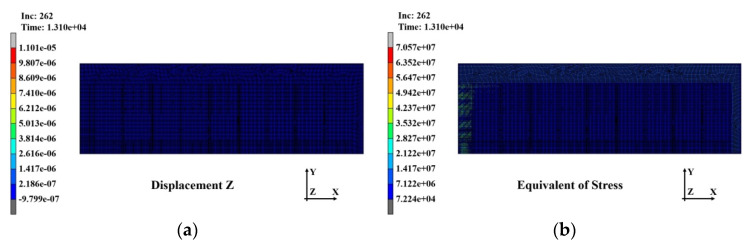
Analysis results of substrate after the third laminating: (**a**) warpage distribution (11.01 μm); (**b**) stress distribution (70.57 MPa).

**Figure 20 polymers-14-00561-f020:**
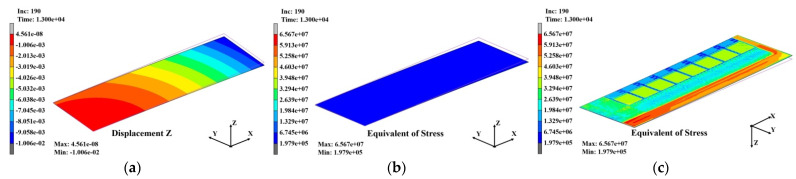
Analysis results of the substrate after SR laminating: (**a**) warpage distribution (10.06 μm); (**b**) stress distribution (65.7 MPa); (**c**) stress distribution at bottom view (65.7 MPa).

**Figure 21 polymers-14-00561-f021:**
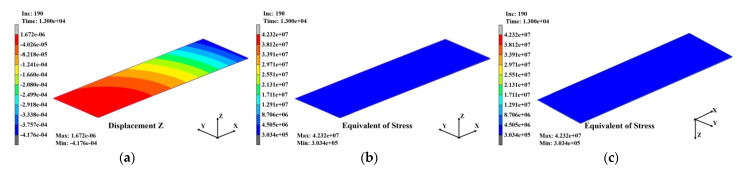
Analysis results of the substrate after SR laminating: (**a**) warpage distribution (0.42 μm); (**b**) stress distribution (42.3 MPa); (**c**) stress distribution at bottom view (42.3 MPa).

**Table 1 polymers-14-00561-t001:** CTE of PPG in an x direction.

Period	Temp.1 [°C]	Temp.2 °C	Dim.1 [μm]	Dim.2 [μm]	CTE [μm/m °C]
1	25	40	−0.10607	0.637419	6.8
2	40	55	0.637419	−0.40103	−9.5
3	55	70	−0.40103	−4.65864	−39
4	70	85	−4.65864	2.056672	62
5	85	100	2.056672	43.81728	380
6	100	115	43.81728	191.7137	1400
7	115	130	191.7137	513.288	2900
8	130	145	513.288	1053.922	5000
9	145	150	1053.922	1092.348	1100
10	150	160	1092.348	1091.73	−85
11	160	260	1091.73	1091.541	−0.26
12	260	25	1093.541	1077.452	94

**Table 2 polymers-14-00561-t002:** CTE of PPG in a y direction.

Period	Temp.1 [°C]	Temp.2 [°C]	Dim.1 [μm]	Dim.2 [μm]	CTE [μm/m °C]
1	25	50	−0.2797	0.5419	45
2	50	65	0.5419	−0.5355	−99
3	65	85	−0.5355	15.0881	110
4	85	100	15.0881	46.1321	280
5	100	110	46.1321	103.1538	780
6	110	120	103.1538	170.5707	930
7	120	160	170.5707	180.27	33
8	160	260	180.27	177.5883	−37
9	260	25	177.5883	162.8827	84

**Table 3 polymers-14-00561-t003:** Material constants of WLF shift function.

Conditions	Direction	$C_{1}$	$C_{2}$
under curing	x	32.3	550.6
	y	8.45	211.5
after cured	x	848.7	2973.1
	y	3003.4	9729.6

**Table 4 polymers-14-00561-t004:** CTE of PPG in a y direction.

Period	Copper	PPG	SR
Elastic modulus [GPa]	80	Ref. Figure 1 and Figure 2	3.5
Poission’s ratio	0.343	0.3	0.3
CTE [μm/m °C]	16.7	Ref. Figure 3 and Figure 4	60
Density [kg/m^3^]	8000	2400	1400
Conductivity [W/m °C]	401	3	0.25
Specific heat [J/kg °C]	385	900	1040

## Data Availability

The data presented in this study are available upon request from the corresponding author.

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
