# Peer review of "Study on Effects of Nonlinear Behavior Characteristics of Prepreg Dielectric on Warpage of Substrate under Laminating Process"

_polymers, 2022, doi:10.3390/polym14030561_

Round 1

Reviewer 1 Report

Study on Effects of Nonlinear Behavior Characteristics of Prepreg Dielectric on Warpage of Substrate under Laminating Process Finite Element Method is used to study the nonlinear behavior of PPG in laminating process. This study is well designed and the authors provide clear numerical evidence to back up their conclusions. In my opinion the quality of this manuscript is high enough for publication, however, the English can be improved.

Author Response

Thank you for your comment.

We reviewed the English language.

Reviewer 2 Report

In this manuscript Cho and Ko have analyzed the effects of nonlinear behavior characteristics of PPG on the warpage of substrate. The authors have done experiment as well as finite element calculations for PPG and SR laminating processes. The results presented in the manuscript are interesting and provide information for minimizing the warpage of substrate in semiconductor chips. I recommend publication of this manuscript subject to the following revisions:

1-Figs. 3 and 4, the unit of dimension change should be micrometer. They read as “um” in the vertical axis. Please fix them.

2-Tables 1 and 2, I assume that columns 4 and 5 in both tables are the dimension change (not the dimension). If so, please correct.

3- Figure 6, is the total volume change (black curve) the sum of volume change by curing (blue curve) and the volume change by CTE (red curve)? If yes, the black curve should indicate the total volume change around 17% at short times. Please make this point clear in the text.

4-A comparison of storage and loss moduli  for x and y dimensions (Figs. 7 and 9) show that for both x and y dimensions the storage modulus is minimum at 150 degree. However the minimum for loss modulus occurs at higher temperatures (200 degree for x and about 175 degree for y). What is the reason for observing the minimum in the loss modulus at a higher temperature in the x dimension, compared to the y dimension?

5-Figures 11 and 12, the scales in the horizontal axis and the numbers are not easily readable. Please update the figure.  

6-Figure 15, the writings in the figure are not easily readable. Please update the figure.

7-The resolutions of Figs. 17-21 are low. Please update the figures.

8-The subsequent heating-cooling cycles in the SR laminating process is reported to decrease the stress inside the substrate. This finding is in close agreement with recent simulation results on two-dimensional layers of colloids in which cooling-heating cycles reduces the stress in the material (J. Chem. Theory Comput. 2021, 17, 1742). May be giving address to this point would be of interest to the readers of this manuscript.

Author Response

Thank you for your comment.

Reviewer 3 Report

Dear Authors,

It was very interesting for me to review an article on such a thorough engineering study. The results obtained are certainly interesting and useful. However, I cannot recommend the manuscript for publication in its current form, due to the following reasons:

1. The introduction should present the current state of the art and indicate relevant references. The reference list contains only a few articles published in the last 3-5 years.

2. It is not entirely clear what the scientific novelty of the proposed approach is. It is required to focus on this attention in the introduction and conclusion.

3. It is necessary to present the results of comparing the proposed approach with existing solutions.

4. The boundaries of the axes in Fig. 10 can be changed to increase the readability of the figures. In the current version, the curves stick together and the difference is not obvious.

5. The grid in Fig. 11 is best removed.

6. Graphs fig. 16 (a, b) can be displayed in the same axes.

7. How was the sampling grid chosen in finite element analysis?

Thus, my decision is major revisions.

Author Response

Thank you for your comment.

Reviewer 4 Report

The manuscript “Study on Effects of Nonlinear Behavior Characteristics of  Prepreg Dielectric on Warpage of Substrate under Laminating Process” analyze the effects of nonlinear behavior characteristics of prepreg (PPG) among the  insulating materials of substrate and the residual stress of laminating process on the warpage of  substrate, this study investigated the continuous laminating process using numerical analysis by  finite element method. The results confirmed that the warpage of substrate is influenced by the solder resist laminating process and that the warpage and stress of substrate accumulated in the laminating process of PPG have significant effects on the final warpage. The manuscript is interesting; however, it is necessary to consider the following observations.

  • Figures 17 to 21 are illegible, using many small figures does not help, it is preferable to select some Figures with higher quality and clarity.
  • Most of the references are not spelled correctly, example 1.S. M. S.; N. S.S.; P. S. H.; K. K.; W. T. Electrophoreti……????

Author Response

Thank you for your comment.

Point 1: Figures 17 to 21 are illegible, using many small figures does not help, it is preferable to select some Figures with higher quality and clarity.

Response 1: They were revised in the paper

Point 2Most of the references are not spelled correctly, example 1.S. M. S.; N. S.S.; P. S. H.; K. K.; W. T. Electrophoreti……????

Response 2: The references have been modified.

Round 2

Reviewer 3 Report

Dear Authors,

Thank you for taking my recommendations into account. I recommend the article for publication in its current form.